# NETWORK PRUNING FOR LOW-RANK BINARY INDEX

## ABSTRACT

Pruning is an efficient model compression technique to remove redundancy in the connectivity of deep neural networks (DNNs). A critical problem to represent sparse matrices after pruning is that if fewer bits are used for quantization and pruning rate is enhanced, then the amount of index becomes relatively larger. Moreover, an irregular index form leads to low parallelism for convolutions and matrix multiplications. In this paper, we propose a new network pruning technique that generates a low-rank binary index matrix to compress index data significantly. Specifically, the proposed compression method finds a particular fine-grained pruning mask that can be decomposed into two binary matrices while decompressing index data is performed by simple binary matrix multiplication. We also propose a tile-based factorization technique that not only lowers memory requirements but also enhances compression ratio. Various DNN models (including conv layers and LSTM layers) can be pruned with much fewer indices compared to previous sparse matrix formats while maintaining the same pruning rate.

## 1 INTRODUCTION

Numerous parameter pruning techniques have been introduced based on the observation that significant amounts of connectivity can be removed without sacrificing model accuracy. Current active research strives to enhance pruning rate at the cost of additional computations (Guo et al., 2016; Molchanov et al., 2017), to reduce computational complexity of pruning procedures (Zhu & Gupta, 2017; Han et al., 2015), and to find the underlying principles of pruning (Liu et al., 2019; Frankle & Carbin, 2019), to name a few.

Figure 1 shows a dense matrix after pruning redundant parameters and various masking index representations. Fine-grained parameter pruning (i.e., each parameter is evaluated to be pruned) results in sparse matrices which can be represented by various formats depending on how indices of non-zero weights are described (Lee et al., 2018a). A desirable index format in sparse DNNs should produce 1) low memory footprint for index and 2) high parallelism for index storage accesses. Unfortunately widely used compressed sparse row (CSR) format involves two index values for each non-zero weight while the number of bits to represent indices increases as the parameter size increases. Even though the binary index form is a regular structure that can utilize full memory bandwidth, sparsity does not reduce memory footprint. A critical concern on those previous sparse matrix formats (CSR format and binary index format) is that the relative amount of index increases if the number of bits to present non-zero weight shrinks due to advanced quantization techniques. Table 1 describes index size divided by the amount of non-zero weights assuming that sparsity is $S$ and non-zero weights are quantized to be $Q$ bits. It is clear that reducing quantization bits aggravates the issue of too large index portion even if a relative indexing scheme (Han et al., 2016b) is used for CSR format with 5 bits.

If pruning is performed in a fine-grained manner to improve overall pruning rate, then each row/column or block exhibits different sparsity. Consequently, row/column-wise or block-wise computations require vastly different computation latency leading to significantly degraded parallelism (Han et al., 2016a). Thus, some of recent pruning techniques suggest removing connectivity in a well-structured form, (Wen et al., 2016; He et al., 2017), or in a block-level (Yu et al., 2017; Narang et al., 2017).

In this paper, we propose a new fine-grained pruning method to find an efficient sparse matrix representation based on *binary index-matrix factorization*. As shown in Figure 1, our proposed index

$$
\begin{bmatrix} 3 & 0 & 0 & 4 \\ 0 & 0 & 0 & 0 \\ 4 & 3 & 0 & 5 \\ 3 & 6 & 0 & 0 \end{bmatrix}
\qquad
\begin{bmatrix} 1 & 0 & 0 & 1 \\ 0 & 0 & 0 & 0 \\ 1 & 1 & 0 & 1 \\ 1 & 1 & 0 & 0 \end{bmatrix}
\qquad
\begin{array}{l} \text{IA} = [\,0\ 2\ 2\ 5\ 7\,] \\ \text{JA} = [\,0\ 3\ 0\ 1\ 3\ 0\ 1\,] \end{array}
\qquad
\begin{bmatrix} 1 & 0 \\ 0 & 0 \\ 1 & 1 \\ 0 & 1 \end{bmatrix}
\begin{bmatrix} 1 & 0 & 0 & 1 \\ 1 & 1 & 0 & 0 \end{bmatrix}
$$

| Dense Matrix after Pruning | Binary Index Format | CSR Index Format | Binary Index Decomposition |

Figure 1: Comparison on various sparse matrix representation formats.

Table 1: Index size divided by weight data size when weights are quantized to be $Q$ bits and pruned to obtain sparsity $S$.

|  | CSR Index | | Binary Index | | | |
|  | 16-bit Index | 5-bit Index | $S$=0.6 | $S$=0.7 | $S$=0.8 | $S$=0.9 |
|---|---|---|---|---|---|---|
| $Q$=1 | 16.0× | 5.0× | 2.5× | 3.3× | 5.0× | 10.0× |
| $Q$=2 | 8.0× | 2.5× | 1.3× | 1.7× | 2.5× | 5.0× |
| $Q$=3 | 5.3× | 1.7× | 0.8× | 1.1× | 1.7× | 3.3× |
| $Q$=4 | 4.0× | 1.3× | 0.6× | 0.8× | 1.3× | 2.5× |

compression scheme decomposes a binary index matrix into two small binary matrices in order to reduce index storage and maintain regular structure. Binary matrix multiplication is inherently parallelizable and sparse non-zero weights can be decoded with high parallelism using recent approaches (Ahn et al., 2019). Thus, decoding sparse matrices can be performed with high performance. In order to accomplish such a new indexing form, we propose an algorithm that finds a particular fine-grained pruning result by generating a low-rank binary index matrix. Since factorization may not exactly reproduce the original binary index matrix, we investigate whether a low-rank binary index matrix produces a pruning result while maintaining allowable model accuracy. Our proposed binary matrix factorization technique significantly reduces the amount of index data compared to using CSR, as demonstrated in the next sections. We also introduce a tiling technique to alleviate on-chip memory size and the burden of on-chip binary matrix multiplication, and to further improve compression ratio.

Recently, a regular-structured index compression method has been proposed using the Viterbi algorithm (Lee et al., 2018a), which explores sequences to be used as pruning mask bits, decompressed by Viterbi encoders. Even though compression rate improves over CSR, for every one input bit, a large number of XOR gates and delay units is required. In comparison, our proposed decompression relies on simple binary matrix multiplications and achieves even higher compression.

## 2 BINARY MATRIX FACTORIZATION FOR PRUNING INDEX

Suppose that a $5 \times 5$ weight matrix $\boldsymbol{W}$ is given as

$$
\boldsymbol{W} = \begin{bmatrix}
-0.1 & 0.9 & 1.2 & -0.2 & -0.6 \\
1.8 & 0.2 & -0.7 & -1.6 & 0.6 \\
-0.1 & -1.7 & 0.1 & -0.3 & 1.2 \\
-0.4 & 1.4 & -0.9 & 0.6 & 1.4 \\
-1.1 & 0.5 & 1.0 & 1.0 & -0.3
\end{bmatrix}. \tag{1}
$$

Following the magnitude-based pruning method (Han et al., 2015), all weights with magnitude smaller than a certain threshold value are pruned to zero. For example, for a threshold of 0.7, we obtain the following pruning index (i.e., binary masking layer) matrix $\boldsymbol{I}$ as

$$
\boldsymbol{I} = \begin{bmatrix}
0 & 1 & 1 & 0 & 0 \\
1 & 0 & 1 & 1 & 0 \\
0 & 1 & 0 & 0 & 1 \\
0 & 1 & 1 & 0 & 1 \\
1 & 0 & 1 & 1 & 0
\end{bmatrix}. \tag{2}
$$

It is important to note that magnitude-based pruning is not an optimal solution (Lee et al., 2018a; LeCun et al., 1990; Han et al., 2016b) to maximize the pruning rate, i.e., a variety of masking layers exist for the same pruning rate.

Binary matrix factorization (BMF) (Zhang et al., 2007) is a reasonable approach to compress $\boldsymbol{I}$. For $\boldsymbol{I} \in \{0,1\}^{m \times n}$, BMF tries to find the best $\boldsymbol{I}_p^{m \times k}$ and $\boldsymbol{I}_z^{k \times n}$ to approximate $\boldsymbol{I}$ as $\boldsymbol{I} \approx \boldsymbol{I}_p \otimes \boldsymbol{I}_z = \boldsymbol{I}_a$, where $k$ corresponds to the binary matrix factorization rank and $\otimes$ stands for binary matrix multiplication. A binary product of $\boldsymbol{I}_p$ and $\boldsymbol{I}_z$ is then defined as

$$(I_a)_{i,j} = \bigvee_{l=1}^{k} (I_p)_{i,l} \wedge (I_z)_{l,j}. \tag{3}$$

While BMF should minimize the number of mismatched bits between $\boldsymbol{I}$ and $\boldsymbol{I}_a$, such an optimization is NP-hard. Hence, several heuristic methods for BMF have been proposed in the literature (Zhang et al., 2007). Moreover, BMF using $\boldsymbol{I}$ (without referring to $\boldsymbol{W}$) lacks weight magnitude information. Yet, in the context of magnitude-based pruning, weights of small magnitude should be pruned with higher probability (i.e., lower importance). Therefore, we explore a method that preserves the importance of each weight when it is not possible to exactly reproduce $\boldsymbol{I}$ through matrix factorization.

## 2.1 BINARY MATRIX FACTORIZATION BASED ON NON-NEGATIVE MATRIX FACTORIZATION

Non-negative matrix factorization (NMF) factorizes a real-valued matrix $\boldsymbol{H}$ into two real-valued matrices $\boldsymbol{H}_1$ and $\boldsymbol{H}_2$ under the constraint that all three matrices consist of non-negative elements. Similar to singular-value decomposition (SVD), NMF attempts to minimize $||\boldsymbol{H} - \boldsymbol{H}_1\boldsymbol{H}_2||_{\mathcal{F}}^2$, where $||\boldsymbol{H}||_{\mathcal{F}}$ denotes the Frobenius norm of the matrix $\boldsymbol{H}$. The property of non-negativity in NMF is useful for analysis of various multivariate data. Numerous numerical approximations of NMF have been suggested since an exact solution of NMF is not generally available (Lee & Seung, 1999; Zhang et al., 2007).

In our proposed technique, we take the magnitude of each element of $\boldsymbol{W}$ to generate $\boldsymbol{M}$ (i.e., $M_{i,j} = |W_{i,j}|$). and $\boldsymbol{M}$ is factorized by an NMF library (e.g., (Zitnik & Zupan, 2012)) into two matrices $\boldsymbol{M}_p$ and $\boldsymbol{M}_z$. For example, a magnitude matrix $\boldsymbol{M}$ of the matrix $\boldsymbol{W}$ of Eq. (1) can be factorized into

$$\boldsymbol{M}_p = \begin{bmatrix} 0.2 & 0.5 \\ 1.3 & 0.0 \\ 0.0 & 0.9 \\ 0.3 & 0.8 \\ 0.8 & 0.2 \end{bmatrix}, \quad \boldsymbol{M}_z = \begin{bmatrix} 1.3 & 0.1 & 0.7 & 1.2 & 0.3 \\ 0.0 & 1.8 & 0.7 & 0.2 & 1.3 \end{bmatrix}, \tag{4}$$

where rank $k$ is 2.

The next step is to convert $\boldsymbol{M}_p$ and $\boldsymbol{M}_z$ into two binary matrices $\boldsymbol{I}_p$ and $\boldsymbol{I}_z$ using threshold values $T_p$ and $T_z$ (i.e., $(I_p)_{i,j} = 1$ if $(M_p)_{i,j} \geq T_p$, or 0 otherwise). The sparsity of $\boldsymbol{I}_p$ and $\boldsymbol{I}_z$ can each be controlled by $T_p$ and $T_z$. Our goal is to achieve similar sparsity between $\boldsymbol{I}$ and $\boldsymbol{I}_p \otimes \boldsymbol{I}_z$. Suppose that $T_p = 0.5$ and $T_z = 0.6$ are carefully chosen to produce similar sparsity as $\boldsymbol{I}$. We obtain

$$\boldsymbol{I}_p = \begin{bmatrix} 0 & 1 \\ 1 & 0 \\ 0 & 1 \\ 0 & 1 \\ 1 & 0 \end{bmatrix}, \quad \boldsymbol{I}_z = \begin{bmatrix} 1 & 0 & 1 & 1 & 0 \\ 0 & 1 & 1 & 0 & 1 \end{bmatrix} \tag{5}$$

and the binary product of $\boldsymbol{I}_p$ and $\boldsymbol{I}_z$ becomes

$$\boldsymbol{I}_a = \boldsymbol{I}_p \otimes \boldsymbol{I}_z = \begin{bmatrix} 0 & 1 & 1 & 0 & \underline{1} \\ 1 & 0 & 1 & 1 & 0 \\ 0 & 1 & \underline{1} & 0 & 1 \\ 0 & 1 & 1 & 0 & 1 \\ 1 & 0 & 1 & 1 & 0 \end{bmatrix}. \tag{6}$$

Compared with the pruning-index matrix $\boldsymbol{I}$ in Eq. (2), there are 2 mismatched elements (underlined in Eq. (6)).

---

**Algorithm 1:** Binary pruning-index-data matrix factorization

---

**input** : $\boldsymbol{W} \in \mathbb{R}^{m \times n}$, rank $k$, target sparsity $S$
**output:** $\boldsymbol{I}_p \in \{0, 1\}^{m \times k}$, $\boldsymbol{I}_z \in \{0, 1\}^{k \times n}$

1: Generate the magnitude matrix $\boldsymbol{M}$ using $\boldsymbol{W}$
2: $\boldsymbol{M}_p, \boldsymbol{M}_z = \text{NMF}(\boldsymbol{M}, k)$
3: $Cost^{min} \leftarrow \infty$, $S_p^{min} \leftarrow 0.0$, $S_z^{min} \leftarrow 0.0$
4: **for** $S_p = 0.0$ **to** $1.0$ **do**
5:     Compute $S_z$ using Eq. (7)
6:     **repeat**
7:        Convert $(\boldsymbol{M}_p, \boldsymbol{M}_z)$ into $(\boldsymbol{I}_p, \boldsymbol{I}_z)$ w/ $(S_p, S_z)$
8:        Adjust $S_z$ depending on $(S_a - S)$
9:     **until** $S_a \approx S$
10:    $Cost \leftarrow \sum\limits_{I_{i,j}=1, (I_a)_{i,j}=0} M_{i,j}$
11:    **if** $Cost^{min} > Cost$ **then**
12:       $Cost^{min} \leftarrow Cost$, $S_p^{min} \leftarrow S_p$, $S_z^{min} \leftarrow S_z$
13:    **end if**
14: **end for**
15: Convert $(\boldsymbol{M}_p, \boldsymbol{M}_z)$ into $(\boldsymbol{I}_p, \boldsymbol{I}_z)$ w/ $(S_p^{min}, S_z^{min})$
16: **Return** $\boldsymbol{I}_p, \boldsymbol{I}_z$

---

The rationale behind this approach is as follows: 1) If $M_{i,j}$ is large, then its corresponding $k$ components of $\boldsymbol{M}_p$ and $\boldsymbol{M}_z$ (i.e., $(\boldsymbol{M}_p)_{i,:}$ and $(\boldsymbol{M}_z)_{:,j}$) will also be large with high probability and, correspondingly, 2) Binary matrix conversion using $T_p$ and $T_z$ would yield a high probability of being '1' within $\boldsymbol{I}_p$ and $\boldsymbol{I}_z$ if the corresponding $M_{i,j}$ is large. Note that in order to match 'OR' operations in Eq. (3), negative values in $M_p$ or $M_z$ are not allowed. Thus, SVD is not applicable for our proposed scheme. Let $S$, $S_a$, $S_p$, and $S_z$ be the sparsity of $\boldsymbol{I}$, $\boldsymbol{I}_a$, $\boldsymbol{I}_p$, and $\boldsymbol{I}_z$, respectively. From the dot product operation in Eq. (6), the expression for pruning rate $S$ becomes

$$S = ((1 - (1 - S_p)(1 - S_z))^k, \tag{7}$$

assuming that the probability of a bit being '0' in $\boldsymbol{I}_p$ and $\boldsymbol{I}_z$ follows $S_p$ and $S_z$. Then, $S_z = (S^{1/k} - S_p)/(1 - S_p)$, which needs to be fine-tuned in practice. If $T_p$ and associated $S_p$ are given, then $T_z$ and $S_z$ are automatically determined by the target pruning rate. Subsequently, given $\boldsymbol{W}$ and rank $k$, it is necessary to find a certain $S_p$ that produces the best masking layer for pruning.

In order to optimize $S_p$, we define the cost function for pruning-index compression to be $\sum M_{i,j}$, where $(I)_{i,j} = 1$ and $(I_a)_{i,j} = 0$ (i.e., a sum of all unintentionally pruned weights' magnitude by binary matrix decomposition). Algorithm 1 describes a method to find the best $S_p$ by sweeping $S_p$ and monitoring $Cost$. Given the linear relationship between $S_a$ and $S_z$, the algorithm can use binary search to expedite adjustment of $S_z$.

## 2.2 MNIST CASE STUDY

We applied our proposed pruning-index compression technique to the LeNet-5 model with the MNIST dataset. LeNet-5 consists of two convolutional layers and two fully-connected layers (Han et al., 2015; Lee et al., 2018b). Since the FC1 layer dominates the memory footprint (93%), we focus on the FC1 layer to investigate the effectiveness of our proposed method, but all layers are pruned with the same rates as in (Han et al., 2015). FC1 involves a $(800 \times 500)$ weight matrix, where full rank is 500. Figure 2 plots $S_z$, $Cost$, and test accuracy across a range of $S_p$ values. Pruning rate $S$ for FC1 is 95% and rank $k$ is 16, 64, or 256. As $k$ increases, both $\boldsymbol{I}_p$ and $\boldsymbol{I}_z$ become more sparse and both $Cost$ and test accuracy improve.

After pre-training for 20K iterations, pruning via BMF is performed as described in Algorithm 1. Then, retraining runs until the 60K[th] iteration using a masking layer $\boldsymbol{I}_a$. The test accuracy (the average of 20 runs) is measured at the 20K[th] (right after pruning), 40K[th], 50K[th], and 60K[th] iterations, as shown in Table 2 along with the compression ratio computed as $mn/(k(m + n))$. Compared with

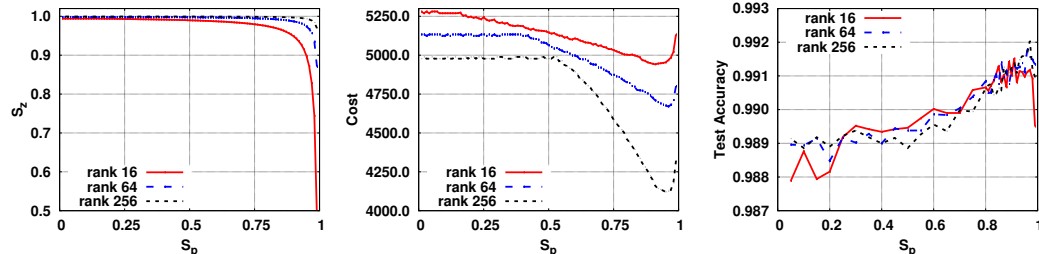

Figure 2: $S_z$, $Cost$, and test accuracy for various $S_p$ when the FC1 layer's pruning-index data of LeNet-5 model is factorized by Algorithm 1 using $S$=0.95, $k = 16$, 64, or 256.

Table 2: (MNIST LeNet-5 accuracy using different rank $k$. At the 20K[th] iteration, we prune all layers using magnitude-based pruning method (Han et al., 2015) except FC1 layer where pruning is performed by using Algorithm 1. Retraining is completed at the 60K[th] iteration. The test accuracy of pre-trained model is 99.2%.

| Rank | Accuracy(%) | | | | Comp. |
|---|---|---|---|---|---|
| $(k)$ | 20K[th] | 40K[th] | 50K[th] | 60K[th] | Ratio |
| 4 | 33.69 | 98.96 | 99.03 | 99.07 | 76.9× |
| 8 | 34.52 | 98.98 | 99.05 | 99.09 | 38.5× |
| 16 | 30.28 | 99.01 | 99.09 | 99.13 | 19.2× |
| 32 | 32.52 | 99.01 | 99.08 | 99.13 | 9.6× |
| 64 | 36.02 | 99.04 | 99.10 | 99.14 | 4.8× |
| 128 | 34.22 | 99.04 | 99.10 | 99.16 | 2.4× |
| 256 | 42.86 | 99.07 | 99.14 | 99.19 | 1.2× |

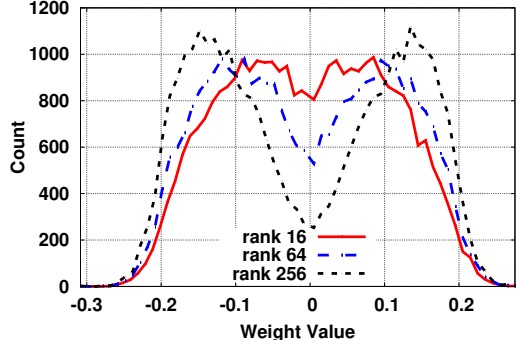

Figure 3: Histogram of unpruned weights of FC1 layer at 20K[th] iteration using LeNet-5 model. Higher $k$ results in more near-zero weights pruned.

other binary pruning index formats (such as ternary quantization), our proposed binary index factorization yields much higher index compression ratios while maintaining reasonable test accuracy.

In general, the histogram of a weight matrix of a pre-trained model follows a Gaussian distribution (Goodfellow et al., 2016). For magnitude-based pruning, the count of near-zero values is significantly reduced after pruning. To investigate which weights are pruned by different rank, Figure 3 presents the histogram of unpruned weight values of the FC1 layer of LeNet-5 model at the 20K[th] iteration right after performing Algorithm 1 using the same pruning rate of 95%. Results show the count of near-zero weights reduces as rank increases. Since the total count is the same in Figure 3 for different rank $k$, a low count of near-zero weights in the histogram implies that fewer weights of large magnitude are pruned. As such, there is a trade-off between the rank $k$ and accuracy even though accuracy drop is reasonable for a wide range of rank values as shown in Table 2. Note that

Table 3: Memory footprint of FC1 layer is compared with various index formats assuming that weights are quantized to be 2 bits (note that index size takes substantial memory footprint for the previous schemes). Accuracy is higher than 99.0% for all formats.

| Method | Index Size | Non-Zero Weights | Sum | Comment |
|---|---|---|---|---|
| Binary | 50.0KB | 5KB | 55KB (100%) | $S$=0.95 |
| CSR(16bit) | 45.8KB | 5KB | 50.8KB (92.4%) | |
| CSR(5bit) | 14.3KB | 5KB | 19.3KB (35.1%) | Relative |
| Viterbi | 10.0KB | 5KB | 15KB (27.3%) | 5X encoder |
| **Proposed** | **2.6KB** | **5KB** | **7.6KB (13.8%)** | $k$=16 |

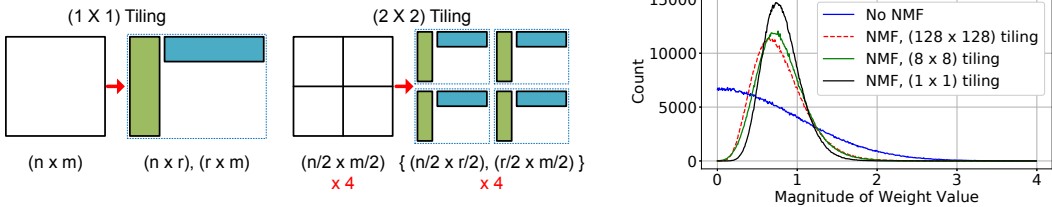

Figure 4: (Left): $(n \times m)$ binary matrix factorization using one or four tiles when the compression ratio is the same. (Right): Weight histogram before and after NMF using different number of tiles.

regardless of the rank $k$, large weights are avoided to be pruned and accuracy is not significantly compromised.

Table 3 compares parameter storage for the FC1 layer using different formats while keeping pruning rate at 95.0%. Reduction in index size for CSR can be achieved using a relative index scheme as described in (Han et al., 2016b). In the case of Viterbi-based pruning, we assume the Viterbi decompressor has 40 outputs, 10-bit comparators, and a skip state of 1 (Lee et al., 2018a). Our proposed pruning method using binary matrix factorization yields the best compression ratio.

## 3 INDEX MATRIX TILING AND WEIGHT MANIPULATION TO LOWER RANK

In this section, we propose two techniques to improve pruning quality for a fixed rank $k$ (or to lower rank $k$ for the same pruning quality).

### 3.1 TILE-BASED BINARY MATRIX FACTORIZATION

Weight matrix size increases quadratically with the number of neurons. If the whole binary matrix multiplication were to be performed within a chip, then on-chip memory size would be prohibitively large. In order to enhance scalability of our proposed factorization method, we propose tile-based factorization as illustrated in Figure 4. A $(n \times m)$ binary matrix is first tiled into multiple blocks. Then, each block is factorized independently with the same rank. Note that tiling size and/or re-shaping the original binary index matrix can be varied to match the underlying parallel computing resources. Because NMF is solved by an iterative method in general, tiling not only reduces the required on-chip memory size but also reduces NMF computation time.

Tiling also affects the quality of binary matrix factorization. Note that when $X_1$, $X_2$, ..., $X_n$ are random samples of size $n$ from a distribution with mean $\mu$ and variance $\sigma^2$, the sample mean $\bar{X}$ shows $E(\bar{X}) = \mu$ and $Var(\bar{X}) = \sigma^2/n$. Indeed, Figure 4 depicts large variance of weight values after NMF when the number of tiles increases (a random Gaussian weight distribution is first assumed). Correspondingly, $M_p$ and $M_z$ also show larger variance with longer tails in the distribution with more blocks as shown in Figure 5. Such increased variance of $M_p$ and $M_z$ with smaller sample size is desirable for binary conversion from the NMF result. For example, compared with $\{0.98, 1.0, 1.2\}$, $\{0.5, 1.0, 1.5\}$ presents larger spectrum of binary conversion threshold values for $S_p$ and $S_z$ (and hence, increases the chance to further optimize the cost function).

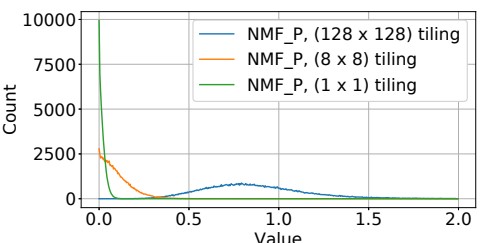 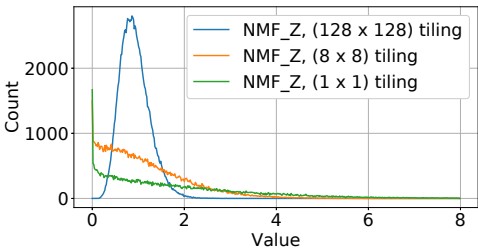

Figure 5: Histogram of $M_p$ and $M_z$ values using different number of tiles.

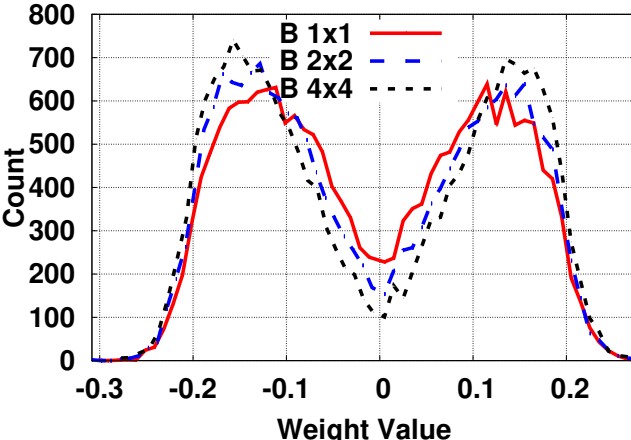

Figure 6: Histogram of unpruned FC1 layer weights of LeNet-5 with different block configurations. Each submatrix is factorized by NMF using different rank depending on the tiling plan to present the same overall compression ratio. Rank 128, 64 or 32 is used for $(1 \times 1)$, $(2 \times 2)$, or $(4 \times 4)$ tiling, respectively.

Figure 6 plots histograms of unpruned FC1 layer weights of the LeNet-5 model on MNIST across different tiling methods. FC1 weight matrix (of $500 \times 800$ size) is tiled into $1(1 \times 1)$, $4(2 \times 2)$, or $16(4 \times 4)$ submatrices while the index compression ratio is the same for all three tiling cases. Since the size of a submatrix differs for each partitioning plan, the rank for factorizing a submatrix is accordingly adjusted in Figure 6. Notice that increasing the number of blocks yields deeper drops of near-zero weights. However, if submatrices have different properties (e.g., an embedding matrix in natural language models) by tiling, each submatrix can optimally choose a different rank.

## 3.2 WEIGHT MAGNITUDE MANIPULATION

If $Cost$ is given as a magnitude-sum of unintentionally pruned weights, then it is still possible that a large weight can be pruned. To prevent large weights from being pruned and to keep the definition of $Cost$, the magnitude of weights can be pre-processed. For example, in the context of magnitude-based pruning, artificially increasing already large weight values can futher lower their probability of being pruned. Note that such weight magnitude manipulation is only temporarily used for pruning-index data compression, and not for normal training or inference steps.

Figure 7 plots histograms of unpruned weights of the FC1 layer of LeNet-5 using different weight-magnitude manipulation methods. In Method 3, weights larger than a threshold are multiplied by $1/(1 - S)$. We observe the sharpest drop of weights around the threshold value and higher count of large weights in Method 3. Finding the best weight manipulation method is an open problem.

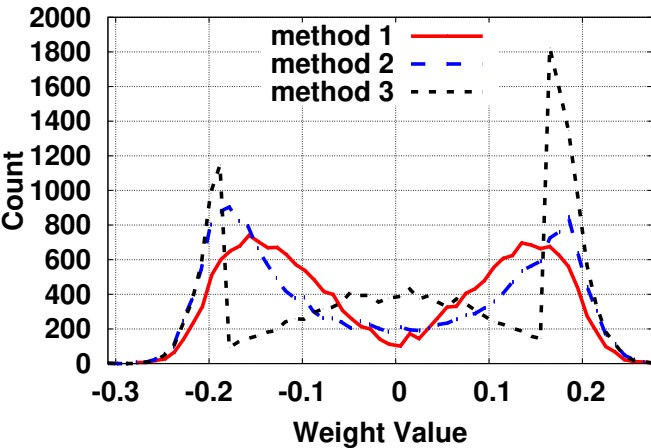

Figure 7: Histogram of unpruned FC1 layer weights of LeNet-5 with different weight magnitude manipulation methods (Method 1: no manipulation, Method 2: $M_{i,j} \rightarrow M_{i,j}^2$, Method 3: $M_{i,j}$ is amplified by $1/(1-S)$ if $M_{i,j}$ is larger than a threshold value (given by a magnitude-based pruning), where $S$ is the sparsity of $\boldsymbol{W}$.

## 4 EXPERIMENTAL RESULTS

We verify our index data compression technique using ResNet32 (He et al., 2016) on CIFAR-10, AlexNet (Krizhevsky et al., 2012) on ImageNet, and a recurrent neural network (RNN) model on the PTB dataset (Marcus et al., 1994), as shown in Table 4. We do not apply binary matrix factorization to layers if their parameter size is relatively small. Such layers usually present small pruning rates (Han et al., 2015; Zhu & Gupta, 2017) and index compression ratio is also small (Yu et al., 2017). For example, Viterbi-based index compression techniques yield 1× compression ratios for the index data of convolutional layers in AlexNet (Lee et al., 2018a).

For ResNet32 mainly consisting of convolutional layers, our pruning-index compression achieves a 70% pruning rate with 91.8% accuracy, providing over 3× index compression as shown in Table 4 (experimental data using various rank selections id provided in Appendix). Note that binary pruning-index matrix decomposition is performed after multiplying the weights (exceeding the pruning threshold) by $1/(1-S)$, as a method to drop more near-zero weights.

For AlexNet, we focus on compressing index data of FC5 and FC6 layers occupying ∼90% of the total model size. Both layers are pruned to a pruning rate of 91% (Han et al., 2015) using Algorithm 1. We achieve over 8× compression in the pruning-index data while maintaining full-precision accuracy. FC5 and FC6 weights are tiled into small blocks, given their large matrix size. BMF of Algorithm 1 is performed on those blocks with ranks 32 and 64, resulting in speed up and reduced $Cost$.

An RNN model with one long-short term memory (LSTM) layer of size 300 (Xu et al., 2018) on the PTB dataset, with performance measured using Perplexity Per Word (PPW), is pruned by our pruning-index-compression method. Note that the embedding and softmax matrices usually take a major memory footprint because of increasing vocabulary size in a neural language model while these two matrices have several distinguished properties compared with general weight matrices (Chen et al., 2018).

To compare the compression ratio of various sparse matrix representation schemes, we choose AlexNet FC5 and FC6 layers and Table 5 shows the index size of a binary matrix representation, CSR format with 16-bit indexing, CSR format with 5-bit indexing (relative indexing as introduced in (Han et al., 2016b)), Viterbi-based representation (Lee et al., 2018a), and our proposed representation. Our proposed network pruning algorithm and index representation using binary matrix factorization significantly reduce the amount of indexing data while maintaining sparsity at the same level as fine-grained pruning. Lastly, while Viterbi-based compression only allows for integer-valued

Table 4: Index compression ratio (compared with binary index scheme) and accuracy of the proposed pruning-index compression method on various DNN models (involving convolutional layers and LSTM layers as well).

| Pre-trained Model | | | Pruned Model using the Proposed Method | | | |
|---|---|---|---|---|---|---|
| Model | # of Weights | Accuracy | $S$ | Rank | Index Data Comp. Ratio | Accuracy |
| ResNet32 on CIFAR-10 | 460.76K | 92.5% | 0.70 | 8/16/32[1] | 3.09× | 91.8% |
| | | | | 8/8/8 | 5.12× | 91.5 % |
| AlexNet on ImageNet (FC5, FC6) | 9K×4K (FC5) | 80.3% (top5) 57.6% (top1) | 0.91 | 32 *tiled*[2] | 8.20× | 80.4% (top5) 57.1% (top1) |
| | 4K×4K (FC6) | | | 64 *tiled*[2] | 4.14× | 80.3% (top5) 57.2% (top1) |
| LSTM on PTB | 6.41M | 89.6 PPW | 0.60 | 145 | 1.82× | 89.0 PPW |

[1] Different ranks are applied to 3 groups of layers according to the number of input channels (16, 32, or 64).
[2] The FC5 and FC6 layers are tiled to 16×8 blocks (of 576×512 size) and 8×8 blocks (of 512×512 size), respectively.

Table 5: Compression results on FC5 and FC6 layers of AlexNet with various index formats when $S$ is to be the same as 0.91 for both layers and non-zero weights are quantized to be 2 bits. Other than the propose scheme, index size is much larger than non-zero weight size. Top-1 accuracy is higher than 57.0% for all formats.

| Method | Index Size | Non-Zero Weights | Sum | Comment |
|---|---|---|---|---|
| Binary | 6656KB | 1198KB | 7854KB (100%) | |
| CSR(16bit) | 10061KB | 1198KB | 11259KB (143.3%) | |
| CSR(5bit) | 3144KB | 1198KB | 4342KB (55.3%) | Relative Indexing |
| Viterbi | 1331KB | 1198KB | 2529KB (32.2%) | 5X Encoder |
| **Proposed** | **812KB** | **1198KB** | **2010KB (25.6%)** | $k$=32, tiled |

compression ratios, our proposed compression technique enables a much wider range of compression ratios (as rational numbers) by controlling the rank for index pruning. Note that more advanced quantization techniques should be supported by stronger index data compression methods. Otherwise, index size would limit the available compression ratio when we combine pruning and quantization schemes.

## 5 CONCLUSION

This paper proposes a fine-grained network pruning technique to produce low-rank binary index matrices. We confirm various DNN models can be pruned by our low-rank indexing scheme while preserving sparsity. Such binary matrix multiplication enables not only high compression ratios, but highly parallel operations of sparse networks. Our proposed tiling method and weight magnitude manipulation schemes further lower rank. Unlike previous studies, this work demonstrates that fine-grained pruning can be represented by a highly regular structured format.

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

## A    SUPPLEMENTARY EXPERIMENTAL RESULTS ON RESNET32 USING CIFAR-10

Table 6 lists the accuracy of ResNet32 using various ranks and pruning rates. The pruned models using our proposed binary matrix decomposition show slightly degraded accuracy compared to the baseline pruning method (the bottom row of Table 6). It can be observed that in general, there exists a trade-off between the ranks and compression ratios.

Table 6: Model accuracy of ResNet32 with CIFAR-10 using different ranks and pruning rates. The bottom row presents the results of baseline pruning method without binary matrix factorization.

| Rank[1] | Index Comp. Ratio | Pruning Rate ($S$) | | |
|---|---|---|---|---|
| | | 0.60 | 0.70 | 0.80 |
| 4/4/4 | 10.29$\times$ | 92.0% | 91.5% | 90.5% |
| 4/8/16 | 6.74$\times$ | 92.2% | 91.5% | 90.9% |
| 8/8/8 | 5.12$\times$ | 92.0% | 91.5% | 90.9% |
| 8/16/32 | 3.09$\times$ | 92.4% | 91.8% | 91.1% |
| 16/16/16 | 2.56$\times$ | 92.2% | 91.8% | 91.0% |
| 16/32/64 | 1.55$\times$ | 92.3% | 91.7% | 91.2% |
| w/o BMF | 1$\times$ | 92.4% | 92.2% | 92.0% |

[1] Different ranks are applied to 3 groups of layers according to the number of input channels (16, 32, or 64).

[2] Retraining is performed for 65K iterations.

