# OpenReview forum: "Network Pruning for Low-Rank Binary Index"
_ICLR.cc/2020/Conference — Reject_

### Official Review · AnonReviewer3 · 2019-10-21
**Official Blind Review #3**

**Rating:** 3

**Review:**

The paper addresses the problem of reducing the computational complexity of neural network pruning. Main idea is to compute a low-rank approximation of the binary index matrix used to represent the structure of the pruned network. In the considered setup, the binary index matrix is the (sparse) boolean matrix associated with the nonzero network's weights. As low-rank decomposition of binary matrices is a hard problem, the authors propose a method to approximate the solution by computing a more standard non-negative matrix factorization.

This is a nice problem and the proposed approach is interesting  but I would tend to reject the paper. The main problem, in my opinion, is a substantial lack of theoretical justifications.  What are the intuitive motivations for considering the proposed method? It is hard to understand why the approach should be preferred to others.  The idea of converting the binary low-rank factorization problem into a real-valued non-negative factorization problem is interesting but
could have been better justified (from an intuitive/theoretical and computational perspective). For example, why is it convenient to convert a hard matrix factorization problem (BMF) into another hard matrix factorization problem (NMF).
An empirical or theoretical analysis of the algorithm's computational complexity would help in this sense.

Also, it is not completely clear what is the relationship of the proposed approach with the neural network framework. Would the impact of the paper increase if presented as a method for binary matrix factorization without links to neural networks pruning? Perhaps a quantitative comparison between gains associated with sparse matrix storage versus other computational costs (related to training or pruning) would help to better collocate the proposed approach in the deep learning framework.

Questions:
- The general idea of approximating BMF with NMF is interesting and could be investigated independently and more deeply. Have the problem and the proposed solution appeared before in the matrix factorization literature (without connections to neural network pruning )?
- Is there any intuitive justification of why thresholding the matrix before and after the factorization leads to consistent results?
- The cost of the proposed approach seems to depend on the rank. Could such explicit dependence be estimated quantitatively and compared with the computational complexity of solving the problem (BMF) directly?

- In the experiments, the value of the pruning objective seems to decrease as the rank of the factorization increases. Is this an expected result? What is even less clear to me is the behaviour of the test accuracy. Is it normal for the pruned network accuracy to increase with sparsity? And why does the performance look independent from the rank?


**Experience Assessment:**

I have read many papers in this area.

**Review Assessment: Checking Correctness Of Derivations And Theory:**

I assessed the sensibility of the derivations and theory.

**Review Assessment: Checking Correctness Of Experiments:**

I assessed the sensibility of the experiments.

**Review Assessment: Thoroughness In Paper Reading:**

I read the paper at least twice and used my best judgement in assessing the paper.

---

### Official Review · AnonReviewer1 · 2019-10-24
**Official Blind Review #1**

**Rating:** 1

**Review:**

This paper proposes an algorithm to compress index data representing the sparsity of a (deep) neural network. The ultimate goal is to reduce the storage requirements for the masks.

The algorithm is based on non-negative matrix factorization so the original mask is decomposed into two smaller matrices leading to memory savings depending on the rank of the matrices.
In addition, the paper proposes a tiling approach to further reduce storage requirements.
Results on selected architectures and datasets show some improvements compared to naive binary index representations

On the positive side, I see an interesting approach to pruning a neural network, assuming the storage cost is guiding the pruning algorithm. If this was integrated into the training process, the optimizer could lead to an optimal solution.


On the negative side, I find the paper not easy to follow/read. A few comments on this regard:

- The motivation is not clear. The paper mostly targets unstructured sparsity (thus the need for sparse matrix representation). However, not really sure if just addressing the storage requirements would have the proper impact. In the end, the original matrix needs to be recovered from the factorization.
- The organization of the paper is also confusing.  Section 2 is about factorization, NMS, then the MNIST case study
- I missed a clear algorithmic section. The title/ conclusions suggest pruning, while the introduction suggests index compression (to me quite different).
Algorithm 1 is for the matrix factorization part but how is this integrated into the entire training / fine-tuning process? That is very confusing to me.

- Interestingly, page 3 starts by suggesting magnitude-based pruning is sub-optimal. However, during the experimental section and the last part of section 2, the paper suggests pruning weights with a large magnitude will damage the performance. That is unclear to me.

- To my understanding, the experiments are not very convincing. Why only those selected architectures and within those pruning only FC layers (AlexNet)? Or, according to table 2 caption, part of the network is pruned using magnitude-based methods and the rest based on the new pruning. That is very confusing. This means the proposal is for pruning or for representing the indexes? If for pruning, why not applying to all the layers in the network? How do we distinguish between the relevance of magnitude pruning and the proposed method?
- How are the different hyperparameters set (S, Sp, k..).
- How does section 3.2 fits with the rest? Methods are vaguely described and then one of them used in the experimental section. Very confusing.
- Table 5 suggests non-zero weights are quantized... I guess I am completely lost now :).

**Experience Assessment:**

I have published one or two papers in this area.

**Review Assessment: Checking Correctness Of Derivations And Theory:**

I assessed the sensibility of the derivations and theory.

**Review Assessment: Checking Correctness Of Experiments:**

I assessed the sensibility of the experiments.

**Review Assessment: Thoroughness In Paper Reading:**

I read the paper at least twice and used my best judgement in assessing the paper.

---

### Official Review · AnonReviewer2 · 2019-10-25
**Official Blind Review #2**

**Rating:** 3

**Review:**

This paper proposed a new network pruning method that generates a low-rank binary index matrix to compress index data, and a tile-based factorization technique to save memory. The binary index can achieve larger compression ratio than the CSR index, and the  low-rank binary index can further reduce memory usage. The results for various networks, including DNN, CNN and LSTM, have shown the effectiveness of the propsoed method. The paper is well-written and easy to follow.

In addition, I have some concerns:
- Discussion about the relationship between your method with binary neural networks [1,2], especially the networks with binary weights [1].
- There are no comparison with the state-of-art methods on pruning and index saving, such as deep compression [3] and CNNPack [4].
- The experiments are not convincing, e.g. only pruning FC5 and FC6 layers in AlexNet on ImageNet dataset.

[1] Courbariaux, Matthieu, Yoshua Bengio, and Jean-Pierre David. "Binaryconnect: Training deep neural networks with binary weights during propagations." Advances in neural information processing systems. 2015.
[2] Hubara, Itay, et al. "Binarized neural networks." Advances in neural information processing systems. 2016.
[3] Han, Song, Huizi Mao, and William J. Dally. "Deep compression: Compressing deep neural networks with pruning, trained quantization and huffman coding." arXiv preprint arXiv:1510.00149 (2015).
[4] Wang, Yunhe, et al. "Cnnpack: Packing convolutional neural networks in the frequency domain." Advances in neural information processing systems. 2016.

**Experience Assessment:**

I have published one or two papers in this area.

**Review Assessment: Checking Correctness Of Derivations And Theory:**

N/A

**Review Assessment: Checking Correctness Of Experiments:**

I assessed the sensibility of the experiments.

**Review Assessment: Thoroughness In Paper Reading:**

N/A

---

### Decision · Program_Chairs · 2019-12-19

**Decision:**

Reject

**Comment:**

The submission proposes a method to improve over a standard binary network pruning strategy by the inclusion of a structured matrix product to encourage network weight sparsification that can have better memory and computational properties.  The idea is well motivated, but there were reviewer concerns about the quality of writing and in particular the quality of the experiments.  The reviewers were unanimous that the paper is not suitable for acceptance at ICLR, and no rebuttal was provided.